# ClutterGen:
# A Cluttered Scene Generator for Robot Learning

**Yinsen Jia**  **Boyuan Chen**
Duke University
http://generalroboticslab.com/ClutterGen

**Abstract:** We introduce **ClutterGen**, a physically compliant simulation scene generator capable of producing highly diverse, cluttered, and stable scenes for robot learning. Generating such scenes is challenging as each object must adhere to physical laws like gravity and collision. As the number of objects increases, finding valid poses becomes more difficult, necessitating significant human engineering effort, which limits the diversity of the scenes. To overcome these challenges, we propose a reinforcement learning method that can be trained with physics-based reward signals provided by the simulator. Our experiments demonstrate that ClutterGen can generate cluttered object layouts with up to ten objects on confined table surfaces. Additionally, our policy design explicitly encourages the diversity of the generated scenes for open-ended generation. Our real-world robot results show that ClutterGen can be directly used for clutter rearrangement and stable placement policy training.

**Keywords:** Simulation Scene Generation, Manipulation, Robot Learning

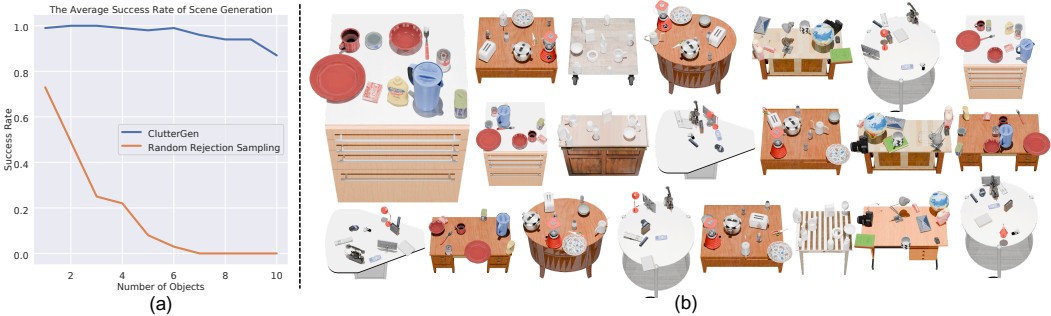

Fig. 1: **(a) The success rate of generating a stable simulation setup.** When the number of objects in the environment increases, the difficulty of creating such a stable setup also increases. The traditional heuristic method cannot create a simulation scene above 7 objects, while ClutterGen consistently achieves high success rates. **(b) Diverse, cluttered, and stable simulation setups created by ClutterGen.**

## 1 Introduction

Simulation has played an important role in advancing robot learning [1, 2, 3, 4, 5] by providing a controlled yet versatile environment for developing and testing algorithms. Data-driven approaches, in particular, typically deploy robots into simulations to undergo extensive training across a variety of diverse and randomized settings to enable generalizable and adaptable behaviors. Significant advancements in robot learning have been achieved by randomizing object shapes [4, 6, 7], textures [8, 9, 10, 11], and dynamics [12]. However, the layout of objects, despite being another critical factor, remains challenging to reach fully open-ended randomization.

Unlike object properties, which can be easily specified within a range without interfering with other objects, object layout must consider the presence of other objects and physical feasibility. For instance, arranging objects in a scene requires ensuring that they do not overlap and are placed in stable positions instead of falling down from the air. Existing efforts often prevent this issue by fixing the object bases [13, 4, 14, 15],

8th Conference on Robot Learning (CoRL 2024), Munich, Germany.

but this strategy is not suitable for many objects like bottles or cups. As the number of objects increases within a limited space, generating a randomized yet stable object layout becomes exponentially difficult. Fig. 1(a) shows the challenge of using the widely adopted approach of random sampling and rejecting failure trials [16, 17, 18, 19] to generate valid scenes, with even seven objects on a table. Other methods require human manual specifications of object regions for local randomization [20, 21, 22] or apply discretization to the possible placement space to avoid collisions [17, 23, 24]. However, navigating and manipulating cluttered environments are essential challenges to deploying robot learning to the real world.

We introduce **ClutterGen**, an auto-regressive simulation scene generator for creating physically compliant and highly diverse cluttered scenes. By framing cluttered scene generation as a reinforcement learning problem, ClutterGen learns a closed-loop policy from 3D observations without requiring pre-existing datasets or human specifications. Once trained, ClutterGen can be applied to variations of the original environment without fine-tuning. We further demonstrate the utility of ClutterGen in several downstream tasks including real-world clutter rearrangement and training robust placement policies for zero-shot sim-to-real transfer.

Our main contributions are summarized as follows:

- We frame the scene generation problem as a reinforcement learning task, enabling a closed-loop policy for creating physically compliant, cluttered, and diverse environments. Our model-free training process uses intuitive physics-based rewards without relying on pre-existing datasets or human specifications.
- Our policy design encourages scene diversity to enhance general robot training.
- We demonstrate ClutterGen's applicability to real-world clutter rearrangement tasks and its effectiveness as a synthetic data generator for training robust stable placement policies.

## 2 Related Work

**Procedural Content Generation (PCG)** Recently, there has been growing interest in data-driven PCG algorithms using generative models for indoor scene generation [25, 26, 27], 3D object asset creation [28, 29], and household rearrangement [30, 31, 32, 32, 33, 34]. Large-language models (LLMs) have also been used to guide scene generation [35, 36, 37] due to their prior embedded human knowledge. However, existing methods often impose strong heuristics such as limiting the regions of each object, and fail to consider physics feasibility, resulting in scenes with intersecting objects, floating objects, and unstable placements that are not suitable for generating flexible, diverse, and realistic robot training environments. In contrast, our method frames scene generation as a reinforcement learning task while explicitly considering the physical stability of the scene generation without relying on human-designed training datasets or expert heuristics.

**Scene-level Randomization for Robot Learning.** Scene-level randomization has been crucial in robot learning. Recent work has focused on synthesizing object shapes [4, 6] and textures [8, 9, 10, 11] to reduce data collection costs and improve model robustness. However, randomizing object poses in simulation scenes remains challenging. Traditional methods, such as random rejection sampling [16, 17, 18, 19], require significant human effort to define the range of object positions and verify scene validity. As the number of objects increases, finding valid poses becomes exponentially harder within a confined region, resulting in low diversity and success rates (Fig. 1(a)). Recently, LLMs have been used to synthesize simulation scenes [38, 22, 39], but they rely on pre-existing datasets or knowledge and extensive user-defined heuristics for object positions. Moreover, LLMs struggle to create accurate object placements without observing physical behaviors, leading to unstable scenes. Our method does not require a high-quality dataset or pre-defined pose constraints and trains by directly interacting with the simulation environment. Our method can create stable, diverse, and highly cluttered scenes for robot training.

## 3 The ClutterGen Framework

Designing a simulation environment for robot training typically involves a human expert observing the scene and object geometry, deciding on a placement, running simulations to check for collision and unstable pose issues, and adjusting placements as needed. This iterative process, repeated until object poses are finalized, is difficult to scale due to its heavy human involvement and time-consuming trial-and-error. We propose **ClutterGen** to automate the steps for generating cluttered simulations using a single learning agent. ClutterGen controls scene generation in a closed-loop manner under challenging cluttered scenarios. Our key idea is to formulate the process as a reinforcement learning task. Without prior knowledge or

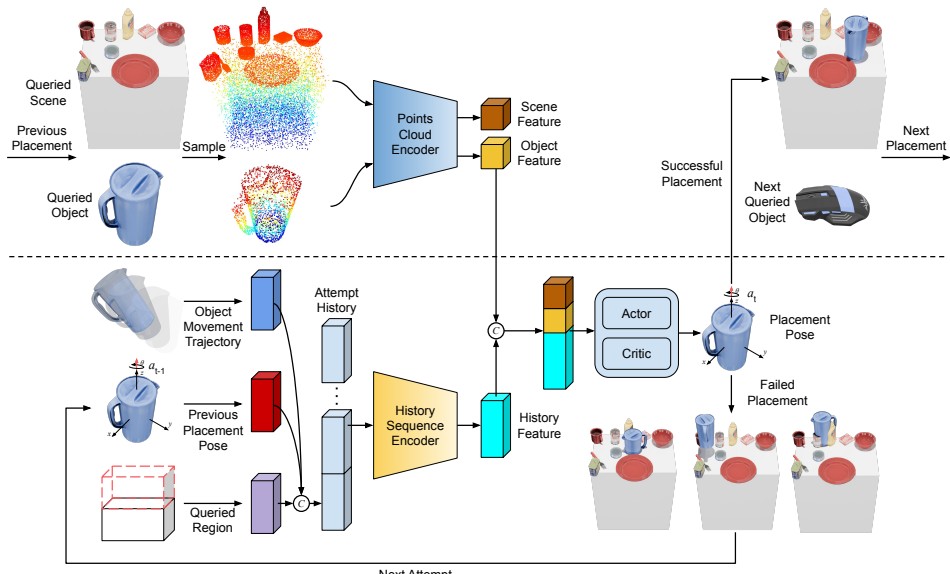

Fig. 2: **ClutterGen.** We stack a sequence of attempt histories as input to the history sequence encoder to generate the history feature. This feature, combined with the perception feature from the point cloud encoder, is taken by ClutterGen to output the placement pose for the queried object. The simulator evaluates the placement's stability, determining whether to proceed to the next attempt or the next queried object placement.

human input, ClutterGen can generate diverse and cluttered scenes using simple reward signals. Fig. 2 shows an overview of our method.

## 3.1 Problem Formulation

Our problem is formulated as follows: A supporting object with a fixed support surface (*e.g.* a table) is placed in the simulation. A queried region is a part of this supporting object, such as a section of a table. The queried scene includes the supporting surface and all previously placed objects. A queried object is the current object to be arranged next. ClutterGen is asked to sequentially place movable objects into the queried region while ensuring stability for the new and previously placed objects. The episode succeeds when all queried objects are placed stably.

We formulate the generation process as a Markov Decision Process, defined by the tuple $<S,A,P,R,\gamma>$. $S$ is a set of states, $A$ is a set of actions, $P$ is a state transition probability matrix; $P(s'|s,a)$ indicating the probability of transitioning to state $s'$ from state $s$ after action $a$, $R$ is a reward function, and $\gamma$ is a discount factor $0 \leq \gamma \leq 1$. A policy $\pi: S \rightarrow A$ determines the next action based on the current state. Our objective is to find an optimal policy $\pi^*$ that maximizes the sum of all discounted future rewards.

## 3.2 Cluttered Scene Generator

**Observation and Action Space** ClutterGen takes in the point cloud of the queried scene and object. If the current object placement fails, we inform the policy about its past actions and their impacts by providing a history of the queried object's movement trajectories $M_i = \{m_0,m_1,...,m_k|m_i \in \mathbb{R}^{13}\}, i = \{1,2,...,I\}$. Each $m$ includes the object's position $\in \mathbb{R}^3$, orientation quaternion $\mathbb{H} \in \mathbb{R}^4$, linear velocity $\in \mathbb{R}^3$ and angular velocity $\in \mathbb{R}^3$ at each step. The hyperparameter $k$ determines the number of simulation steps to verify placement success, and $I$ is the predefined maximum number of attempts allowed.

We encode each point cloud with a feature encoder to extract geometry embeddings. We concatenate the movement trajectory, previous placement poses, and queried regions and send them into a history sequence encoder to generate a history embedding. The concatenation of geometry and history embedding is the final input to our RL agent. Our policy outputs the object placement position and z-axis rotation relative to the queried region.

**Policy Design** We optimize our RL agent with PPO [40] which is an on-policy actor-critic algorithm. The critic computes the advantage during training and the actor generates the action distribution for the 3D

translation and 1D rotation independently. Since our design requires bounded continuous action space, instead of using the conventional squashed normal distribution to bound the action space, we choose beta distribution as the policy distribution. Specifically, the actor of the agent outputs the $\alpha \in \mathbb{R}^4$ and $\beta \in \mathbb{R}^4$ to build the action distribution $Beta(\alpha,\beta)$, and then the relative placement pose can be obtained by sampling from the distribution.

Our choice of action distribution offers several benefits. First, beta distribution is well-defined among $[0,1]$, which only requires a simple linear operation to our action range $[-1,1]$, while the unsquashed process for the squashed normal distribution will introduce numerical instability during training and requires extra post-processing. Second, beta distribution can represent more distribution shapes by changing the $\alpha$ and $\beta$ values, which is essential to improve the diversity of our scene generation. As shown in Sec. 4.1, our results demonstrate that the policy trained with the beta distribution outperforms the policy trained with the squashed normal distribution in both success rate and scene diversity.

**Success Conditions** A successfully placed object should not collide with existing objects and maintain a stable pose. To verify a placement, we run the simulator and record the object's velocity and acceleration to check stability. The placement is considered stable if the following conditions are met: 1) the linear velocity and acceleration become less than $0.005$m/s and $1$m/s$^2$, respectively, within 40 simulation steps (0.167s in real-time); 2) the angular velocity and acceleration become less than $0.5°$ and $180°/s^2$, respectively, within 40 simulation steps; and 3) the velocity and acceleration remain below these thresholds for 20 continuous simulation steps.

**Reward Function** Guided by the success conditions, for each placement attempt, ClutterGen optimizes the following reward function to minimize the accumulated absolute values of the velocity and accelerate during new object placements:

$$R_i = -c\sum_{i=0}^{k}(||v_i||_2 + ||a_i||_2) + n \cdot \mathbb{1}_{\text{stable}} \cdot R_0 \tag{1}$$

where $c$ is a scaler to adjust the velocity and acceleration penalty, $v_i \in \mathbb{R}^6$ and $a_i \in \mathbb{R}^6$ represents the velocity and acceleration the queried object at $i_t h$ simulation step, the indicator function $\mathbb{1}_{\text{stable}}$ will equal to 1 if the placement pose is stable otherwise will equal to 0, $n$ represents the current queried object is the $n_{th}$ object for the queried scene, and $R_0$ is a scalar reward.

### 3.3 Implementation Details

For each attempt, we run the simulator for up to $k=240$ steps (1s) to check the placement stability. We set the maximum allowed attempts as $I=5$. We use PointNet++ [41] as the point cloud feature encoder and 3-layer MLP as the history sequence encoder. Both the actor and critic networks are composed of 5 layers of MLPs. We optimized the agent for 2.5M steps with a learning rate of $1e^{-4}$. All training was conducted on one NVIDIA RTX A6000 GPU. Other details are listed in the Appendix.

## 4 Experiments

In this section, we aim to evaluate ClutterGen's capability to effectively generate cluttered, stable, and diverse scenes. We further investigate the generalizability of the learned model on various scene-level changes. We then conduct several real-world experiments to demonstrate the effectiveness of ClutterGen for downstream robotics tasks such as clutter rearrangement and stable placement policy training.

### 4.1 Scene Generation Evaluation

**Dataset** To include diverse everyday objects for our evaluation, we created a dataset consisting of five groups, each containing ten objects. The first four groups are selected from the PartNet-Mobility [42], Objaverse [43], and YCB [44] datasets. The fifth group, referred to as the *real group*, includes our 3D-scanned everyday objects that are also used in our physical experiments. We used a cuboid table with $[60cm,70cm,70cm]$ in width, length, and height as the initial queried scene.

**Baselines** Since most prior studies on scene generation rely on either pre-collected datasets or human-specified heuristics and constraints, we cannot directly compare with them. The closest method, which is also widely adopted [16, 17, 18, 19] in recent literature, is the random rejection sampling (RRS) algorithm. We also compare our methods with other alternative settings.

| Method | Group 1 Success Rate ↑ | Group 1 Stable Steps ↓ | Group 2 Success Rate ↑ | Group 2 Stable Steps ↓ | Group 3 Success Rate ↑ | Group 3 Stable Steps ↓ | Group 4 Success Rate ↑ | Group 4 Stable Steps ↓ | Group 5 Success Rate ↑ | Group 5 Stable Steps ↓ |
|---|---|---|---|---|---|---|---|---|---|---|
| RRS | 0.005 | 168.9±202.1 | 0.00 | 290.3±379.1 | 0.00 | 171.3±188.2 | 0.02 | 214.2±275.8 | 0.005 | 175.6±198.2 |
| ClutterGen-OL | 0.212 | 139.4±168.2 | 0.145 | 206.7±247.5 | 0.086 | 170.9±179.4 | 0.232 | 164.9±182.1 | 0.251 | 135.0±121.1 |
| ClutterGen-SM | 0.359 | 101.8±125.1 | 0.414 | 111.3±149.5 | 0.364 | 106.6±119.3 | 0.517 | 97.62±101.8 | 0.602 | 83.47±90.26 |
| ClutterGen-Normal | **0.925** | **73.86±57.73** | 0.833 | 85.25±91.44 | 0.951 | **56.37±61.13** | 0.969 | 51.19±56.22 | **0.989** | **43.86±33.40** |
| ClutterGen | 0.912 | 88.50±95.46 | **0.874** | **70.82±97.86** | **0.963** | 59.70±55.73 | **0.973** | **49.89±55.55** | 0.988 | 45.72±31.21 |

Tab. 1: **Success rate comparison of cluttered scene generation.** We report the average success rate and stable steps across three random seeds of training and five object groups. Our method significantly outperforms the widely adopted RRS baseline. The long-term attempt history and closed-loop design in our framework deliver the best performance.

- *Random Rejection Sampling (RRS).* This method heuristically computes the position of the supporting surface in the queried scene and randomly places objects within the queried region.
- *ClutterGen-OpenLoop (OL).* This method uses the same architecture as ClutterGen, but without previous object movement trajectory and placement pose information for the next attempt.
- *ClutterGen-ShortMemory (SM).* This method uses the same architecture but only takes the latest attempt history (buffer length = 1) for the next attempt.
- *ClutterGen-Normal.* This method uses the truncated normal distribution instead of the beta distribution.

**Metrics** Our evaluation metrics include: 1) the *success rate* of placing all queried objects into the queried scene, and 2) the average simulation steps (*stable steps*) required for each object to achieve stability. Additionally, we access *setup diversity*, a qualitative metric, using a diversity map to evaluate the variety of the generated scenes.

**Results** Tab. 1 shows the testing restuls with 1,000 trials. The RRS almost always failed to produce a valid object layout. This is because RRS cannot learn from active interaction with the environment. While ClutterGen-OL and CluuterGen-SM can produce some reasonable layouts, the success rate is generated low with longer simulation steps for the scene to become stable, suggesting long-term close-loop history is important. Both the normal and beta distributions achieve high success rates.

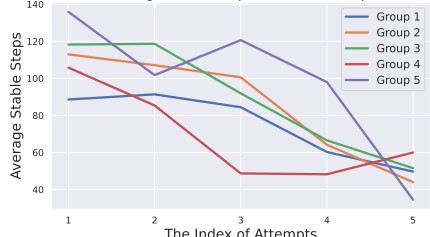

Fig. 3: **Average stable steps across different numbers of attempts.** We computed the average stable steps for object placements requiring ≥3 attempts. The x-axis represents the $i_{th}$ attempt, and the y-axis represents the average simulation steps for the object to stabilize. ClutterGen's closed-loop re-attempt mechanism increasingly improves placement stability.

To further understand the effectiveness of ClutterGen, we plot the average stable steps needed over different attempts. As shown in Fig.3, ClutterGen can use previous placement poses and object movement trajectories to adjust the next placement pose for faster convergence of stability. This closed-loop mechanism is extremely useful for cluttered scene generation within limited spaces. Fig. 4 shows an example of ClutterGen's closed-loop generation process to adjust its next action based on the past attempts.

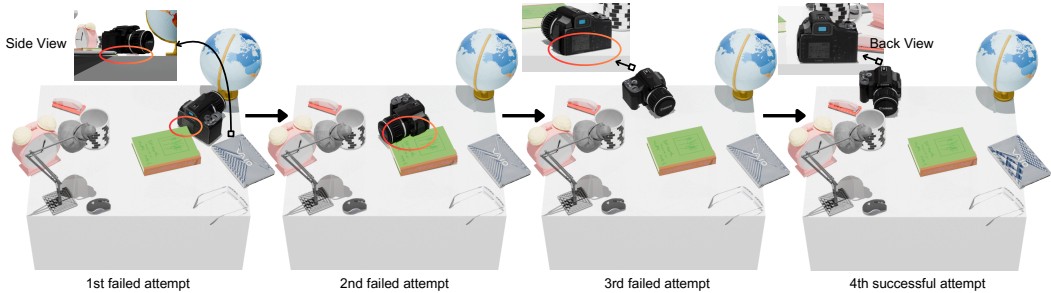

Fig. 4: **Example of the closed-loop generation process of ClutterGen.** ClutterGen attempts to place a camera on a cluttered table. Failed placement attempts (e.g., floating in the air, colliding with objects, or inserting into the table) are marked by red circles. After each attempt, the simulator runs and records the movement trajectory. ClutterGen uses all previous failed attempts to adjust its future actions until achieving a successful placement.

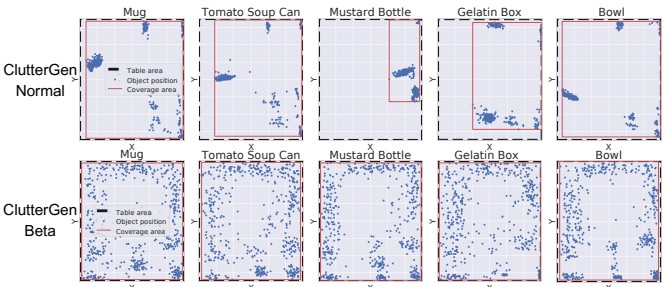

Fig. 5: **Generation diversity.** A projected view of the queried object's placements. The black dashed line represents the supporting surface area. The blue dots are queried object placement positions $(x,y)$ across 500 setups. The red box is the coverage area, bounding all placement positions. Beta distribution greatly enhances scene diversity.

| **Method** | *Original* | *Translation* $x$:$[-15cm,15cm]$ $y$:$[-15cm,15cm]$ | *Rotation* $r_z$:$[-\pi,\pi]$ | *Shrinkage* $\Delta h_x$:$[-10cm,0cm]$ $\Delta h_y$:$[-10cm,0cm]$ | *Expansion* $\Delta h_x$:$(0cm,10cm]$ $\Delta h_y$:$(0cm,10cm]$ | *Randomly Combined* |
|---|---|---|---|---|---|---|
| ClutterGen | 0.99 | 0.89 | 0.92 | 0.76 | 0.93 | 0.85 |

Tab. 2: **Scene-level generalization results.** We report the average success rate of cluttered scene generation under various test-time changes to the queried region. ClutterGen demonstrates strong generalizability.

Generating diverse object layouts is crucial for training robust robot policies by providing sufficient randomization. Although both ClutterGen and ClutterGen-Normal achieved high success rates in generating cluttered scenes, we assess their ability to create diverse layouts by visualizing their successful placements through a 2D projection. Each subplot in Fig. 5 shows the placement distribution for one object across 500 scenes. ClutterGen-Normal tends to place objects in very similar positions resulting in homogeneous outcomes, while our ClutterGen with beta distribution generates significantly more diverse setups.

## 4.2 Scene-level Generalization

We are interested in evaluating ClutterGen's generalization capability to generate cluttered scenes when the queried region varies during test time, even if it was fixed during training. We propose five test-time changes: 1) translation (random 2D translation within a certain range), 2) rotation (random rotation around $z$-axis), 3) shrinkage (random reduction of $xy$ half-extents), 4) expansion (random enlargement of $xy$ half-extents), and 5) randomly combinations of the previous changes. We used the real object group for this evaluation. To ensure the queried region remains covered by the support surface after the changes, we enlarged the table dimension to

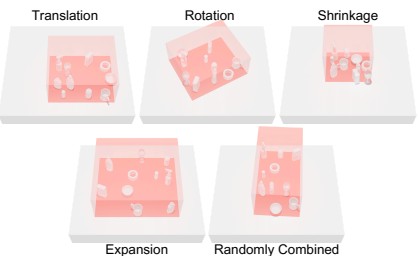

Fig. 6: Qualitative examples of the generated scenes under test-time changes of the queried region.

$[140cm,140cm,70cm]$ for this test. Our evaluation across 500 episodes for each change is shown in Tab. 2. We observe a performance drop in the shrinkage test. This is because a narrower queried region forces the scene to be more cluttered, which increases the difficulty of finding stable poses for all objects. Overall, ClutterGen shows strong zero-shot generalization ability across all changes. Examples of the generated scenes are shown in Fig. 6.

To demonstrate the real-world use cases, we selected 10 furniture tables from our dataset and directly applied ClutterGen to them with all five object groups. We adjusted the queried region's $xy$ extents to match the furniture's dimension. We also randomly applied the above four changes. By evaluating 50 episodes for each table, we achieved an overall 70% success rate. Qualitative examples are shown in Fig. 1. Another advantage of our method is the formulation of scene generation as a sequential 3D object placement task. Therefore, ClutterGen can naturally generate complex object relationships such as *a mug on a book* or *a fork under a plate*.

## 4.3 Real Robotics Task: Clutter Rearrangement

Given a clutter of objects, humans can easily determine the goal state of stable poses for each object when tasked with moving them to another location (e.g., from one table to another). However, achieving this behavior in robots remains challenging. For instance, when the target area is cluttered, robots must identify a safe and stable pose for new object placements to avoid collisions or simply dropping objects into the area. Consequently, existing "pick-and-drop" solutions are inadequate in this scenario. Additionally, human

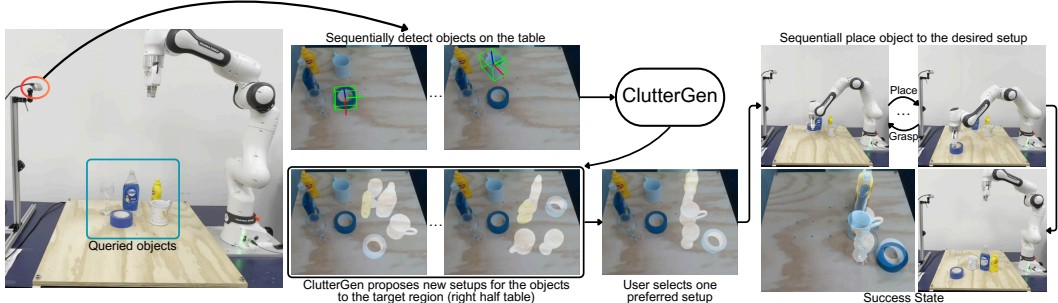

Fig. 7: **Clutter rearrangement.** The red circle indicates the position of the camera. The blue cube encloses the objects to be rearranged. ClutterGen proposes several different setups for rearrangement. After a user selects a preferred setup, the Panda arm will rearrange the table accordingly.

preferences for the resulting scene layout can vary, making it difficult to incorporate these preferences into robot planning.

We demonstrate ClutterGen's applicability in the challenging task of clutter rearrangement using a Franka Panda arm equipped with an RGB-D camera. Fig. 7 provides an overview of our approach. We randomly selected five objects from our real object group and placed them on one side of a table as the initial clutter. The robot's goal is to move this entire clutter to the other side of the table. We used GroundDINO [45] to identify the object categories, Segment Anything [46] to generate the object masks, and FoundationPose [47] to determine the 6 DoF pose of each object. To simplify the setting and focus on ClutterGen's key capabilities, we assume access to the 3D models of these objects and a set of grasping poses [48], although these assumptions can be relaxed with recent advancements in grasp synthesis [49, 50, 51]. With this information, we render the queried scene point cloud to ClutterGen to propose possible target layouts.

We asked ClutterGen to generate ten possible layouts for the other side of the table, allowing users to select their preferred setup and obtain the target poses for each object. The robot arm then planned its motions to move the entire clutter to the target area using MoveIt [52]. We evaluated the full pipeline over ten episodes. Success is counted if all objects are rearranged to the user-selected setup without any collisions or unstable poses. The overall success rate was $7/10$. We include both successful and failed trials in the supplementary videos. Failures were due to arm-object or object-object collisions during motion planning (2/3) or failed object layout proposals from ClutterGen under the ten-trial limit (1/3).

### 4.4 Real Robotics Task: Stable Object Placement

While pick-and-drop tasks have seen great success in robot manipulation, placing objects stably in cluttered scenes remains challenging. In this experiment, we leverage ClutterGen as a synthetic data generator to train a robot policy for stable object placement. Fig. 8 provides an overview of our approach. Our stable

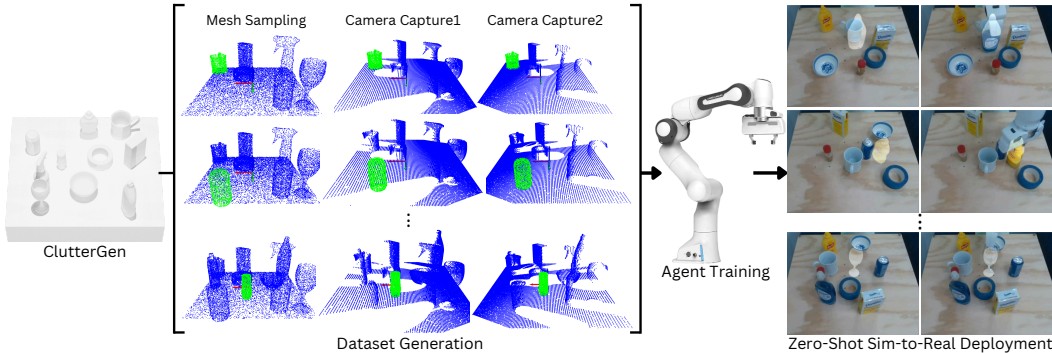

Fig. 8: **Synthetic dataset generation and zero-shot sim-to-real policy deployment.** We generate a synthetic dataset by replaying the scene generation trajectory created by ClutterGen. Green points mark the ground truth pose of the queried object, while blue points represent the point cloud of the queried scene. To augment the dataset, we use a virtual camera to capture the scene point cloud from different angles. This synthetic dataset is then used to train a stable object placement policy, which is directly deployed on a real robot.

placement policy takes the point clouds of the queried scene and object and learns to output the stable placement pose. Due to our novel formulation of ClutterGen as a sequential 3D placement task, ClutterGen naturally serves as a data generation source for training the stable placement policy. Specifically, we ran ClutterGen 500 times to create 500 layouts while recording its generation process for each object, resulting in 5,000 data points. To mitigate occlusion issues in the real world, instead of using the full point cloud, we captured the scene point cloud using a virtual RGB-D camera in the simulator and randomly translated its position four times while ensuring focus on the center of the scene. We then randomly sampled the final 5,000 data points as our dataset termed *ClutterGen-5000*. This data generation process takes less than **one minute** on a single thread and can be further accelerated through parallelization. We then applied supervised learning to train the stable placement policy over 1,000 epochs using mean squared error loss. At test time, we achieve a $91\%$ success rate by using the simulator to verify placement success with the same physics-based stability metric.

To demonstrate the efficiency improvements of ClutterGen compared to a human engineer, we developed a user interface within the Pybullet [53] simulator to allow humans to manually create simulation setups. We manually constructed 20 stable setups, resulting in 200 data points. We applied the same augmentation process as above to obtain the final dataset termed *Human-200*. The entire process took approximately 1.5 hours for an expert

| Dataset | ClutterGen-5000 | Human-200 |
|---|---|---|
| Success Rate | 0.91 | 0.52 |

Tab. 3: **Boost model performance by training with the synthetic dataset from ClutterGen.** We report the success rate of two models trained on *ClutterGen-5000* and *Human-200*. ClutterGen can create diverse and effective training data in under one minute, whereas a human expert needs 1.5 hours to create a dataset that is only $4\%$ of its size.

engineer. We then trained the stable object placement policy using this dataset under the same configurations as before. Both models were evaluated on the same test dataset as shown in Tab. 3. *ClutternGen-5000* enables significant higher performance than using *Human-200* for the policy training.

We directly deployed the stable object placement policy trained with *ClutterGen-5000* to a Franka Panda arm equipped with an RGB-D camera. In each episode, the robot arm grasped and lifted a queried object while six to eight other objects were randomly placed on the table to form an unseen clutter. Based on the scene point cloud captured by the RGB-D camera and the object point cloud, the policy predicted a stable placement pose, which was then used to plan and execute the robot's actions. We repeated this process by randomizing the clutter configurations for ten episodes per object. We used five target objects to be placed. An episode was counted as successful if the object was stably placed without any collisions. Failures occurred if the model failed to predict a stable pose, the arm could not reach the pose after five attempts, or a collision occurred during placement. Across all 50 episodes, our zero-shot sim-to-real policy achieved a $72\%$ success rate. The performance drop is likely due to significant noise in the real-world point cloud. This could be mitigated by further randomizing the dataset to simulate real-world noises or using better hardware to capture the point clouds.

## 5 Conclusion, Limitations, and Future Work

In this work, we propose ClutterGen, an auto-regressive simulation scene generator for robot learning. ClutterGen efficiently generates diverse, cluttered, and physically compliant environments without relying on pre-existing datasets or human specifications. Our key idea of framing the scene generation task as a reinforcement learning problem enables a closed-loop mechanism to sequentially generate 3D object placements. Our policy design further enhances the diversity of the generated scenes. Through both simulation and real robot experiments, we demonstrate that ClutterGen can help tackle several challenging robotics tasks, such as clutter rearrangement and stable object placement in cluttered environments.

Our work has a few limitations that should be addressed in future research. First, the object layouts are currently limited to rotation along the $z$-axis. Allowing rotations along the $x$ and $y$ axes could produce more diverse and complex layouts. Second, we have only demonstrated the setups on flat surfaces. Future work could explore various surface configurations, such as uneven or inclined surfaces. Finally, our current training involves only ten objects in a limited space. Training the policy with a great number of objects in the pool could enhance object-level generalization during testing. Overall, we hope that ClutterGen provides a novel problem formulation for automated simulation scene design to facilitate the generation of large datasets for robot learning.

**Acknowledgments**

This work is supported by ARL STRONG program under awards W911NF2320182 and W911NF2220113, by DARPA FoundSci program under award HR00112490372, and DARPA TIAMAT program under award HR00112490419.

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

## Appendix

**Loss Function** We use the PPO algorithm to optimize CSG. The policy objective function is given by

$$L^{\text{CLIP}}(\theta)=\mathbb{E}_t\left[\min\left(r_t(\theta)\hat{A}_t,\text{clip}(r_t(\theta),1-\epsilon,1+\epsilon)\hat{A}_t\right)\right] \tag{2}$$

$$r_t(\theta)=\frac{\pi_\theta(a_t\,|\,s_t)}{\pi_{\theta_{\text{old}}}(a_t\,|\,s_t)} \tag{3}$$

where $r_t(\theta)$ is the probability ratio between the new policy and the old policy; $\hat{A}_t$ is the advantage function estimate at timestep $t$; $\text{clip}(\cdot)$ is a function that clips $r_t(\theta)$ within the range $[1-\epsilon,1+\epsilon]$; $\epsilon$ is a small hyper-parameter that controls the clipping range.

The value function error and the entropy bonus are given by

$$L^{\text{VF}}(\theta)=\mathbb{E}_t\left[\left(V_\theta(s_t)-\hat{R}_t\right)^2\right] \tag{4}$$

$$L^{\text{H}}(\theta)=\mathbb{E}_t[\mathcal{H}[\pi_\theta](s_t)] \tag{5}$$

where $V_\theta(s_t)$ is the value function parameterized by $\theta$; $\hat{R}_t$ is the cumulative return at timestep $t$; $\mathcal{H}[\pi_\theta](s_t)$ is the entropy of the policy at state $s_t$.

The combined loss function is given by

$$L(\theta)=c_1 L^{\text{VF}}(\theta)-c_2 L^{\text{H}}(\theta)-L^{\text{CLIP}}(\theta) \tag{6}$$

where $c_1$ and $c_2$ are the coefficients that balance the importance of the value function error and the entropy bonus, respectively.

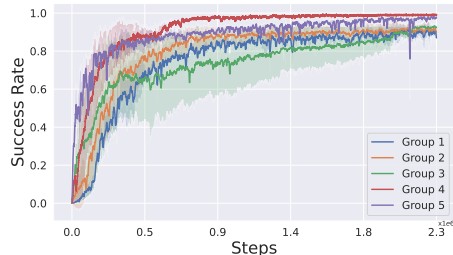

Fig. 9: **Learning curves of ClutterGen.** The x-axis shows the number of steps, and the y-axis shows the success rate. The solid curve represents the average success rate from three different initial random seeds for each group, with the semi-transparent area indicating the standard deviation.

Fig. 10: **Maximum allowed attempts selection.** The performance will drop if the maximum allowed attempts $\leq 2$. We choose 5 attempts for the ClutterGen.

**Extra Explanation of the Stable Conditions** It is difficult to precisely define object stable conditions with single thresholds for the velocity and acceleration in the simulation. On one hand, the simulator is not perfect, meaning a visually stable pose usually has non-zero velocity and acceleration. Too small thresholds for the velocity and acceleration usually introduce false negatives (*i.e.*, the queried object that is already stable by human judgment but can not pass the stable conditions). On the other hand, too large thresholds usually introduce false positives. In this work, we applied small thresholds for both values, formulating a relatively strict checking condition.

**Implementation Details of the ClutterGen** The PointNet++ encoder was pre-trained on modelnet40 dataset [54] and the weights were frozen during the training of ClutterGen. The attempt history was flattened and then padded by 0 if the current attempts were fewer than the maximum allowed attempts to

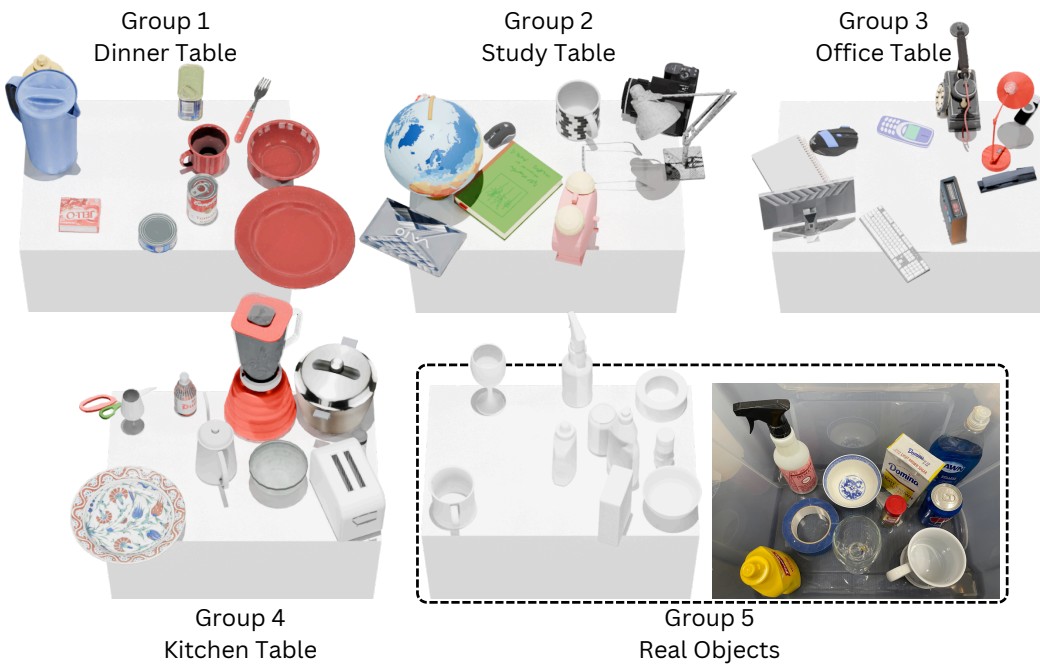

Fig. 11: **5 groups objects dataset.** Each group contains 10 objects. The group 1 to 4 are selected from the existing object dataset. Group 5 is 3D-scanned everyday objects that are also used in our physical experiments.

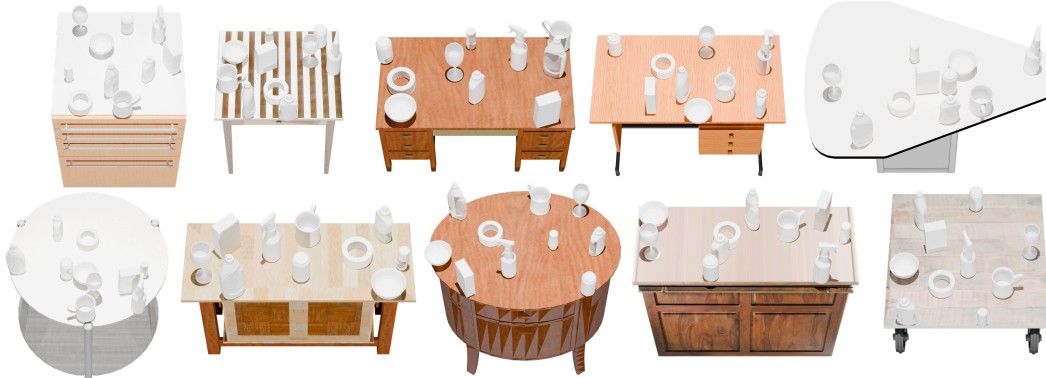

Fig. 12: **10 test tables dataset.** We select 10 different tables from the existing object dataset for the scene-level randomization test.

guarantee fixed-size input before being sent to the history sequence encoder. All the training and test was conducted in the PyBullet simulator [53].

**ClutterGen Training Results** We trained ClutterGen on different group objects. The training results are shown in Fig.9. ClutterGen could reach above 0.85 success rate after 2.3M steps optimization for all groups, which shows the high adaptation of different objects.

**Selection of Maximum Allowed Attempts.** We test the performance of ClutterGen with different maximum attempts on group 1 objects. Increasing allowed attempts will boost the performance. However, too many attempts will slow down the training process. We set 5 attempts at most to balance the performance and training speed.

**Implementation Details of Stable Placement Policy** We used two pre-trained PointNet++ models as the perception encoders for the queried scene point cloud and the queried object point cloud respectively. We did not freeze their weights during the training in this task. The perception features were concatenated and

sent to a 4-layer MLP to generate the predicted placement pose into the scene surface center frame. This pose was then transformed into the robot base frame for motion planning.

Tab. 4: PPO Hyperparameters

| Hyperparameter | Value | Description |
|---|---|---|
| Learning Rate | 0.0001 | The step size at each iteration while moving toward a minimum of a loss function. |
| Batch Size | 1000 | Number of steps per gradient update. |
| Update Epochs | 5 | Number of epochs to update the policy. |
| $\gamma$ | 0.99 | The discount factor for reward. |
| $\lambda_{gae}$ | 0.95 | The discount for the general advantage estimation. |
| $\epsilon$ | 0.2 | The surrogate clipping coefficient. |
| $c_1$ | 0.5 | The coefficient of the value function. |
| $c_2$ | 0.01 | The coefficient of the entropy. |
| $g_{clip\_max}$ | 0.5 | The maximum norm for the gradient clipping. |
| Hidden Layer Size | 256 | The number of units of hidden layer in MLP. |
| Optimizer | Adam | Optimization algorithm for updating weights. |

Tab. 5: ClutterGen Hyperparameters

| Hyperparameter | Value | Description |
|---|---|---|
| $P_{obj}$ | 1024 | Number of points of queried object point cloud. |
| $P_{sce}$ | 20480 | Number of points of queried scene point cloud. |
| Max Attempts | 5 | Number of allowed attempts of placing the queried object. |
| $c$ | 0.005 | The coefficient of the velocity and acceleration penalty. |
| $R_0$ | 100 | The scalar reward. |

