# OpenReview forum: "ClutterGen: A Cluttered Scene Generator for Robot Learning"
_robot-learning.org/CoRL/2024/Conference — CoRL 2024_

### Official Review · Reviewer_qZ9R · 2024-07-20
**ClutterGen: A Cluttered Scene Generator for Robot Learning**

**Originality:** 3
**Technical Quality:** 3
**Clarity Of Presentation:** 5
**Potential Impact:** 3
**Recommendation:** 2
**Confidence:** 4

**Review:**

**Summary**

The paper presents a novel approach to generating physically compliant, cluttered scenes for robot learning using reinforcement learning.  The presented framework framework that casts the scene generation problem as a reinforcement learning task, using a closed-loop mechanism and beta distribution for action space to enhance scene diversity and stability.  The paper provides extensive evaluation through simulations and real-world experiments, demonstrating the effectiveness of ClutterGen in generating diverse and stable object layouts.

**Strengths**

- One of the biggest strengths of the paper is the thorough evaluation of ClutterGen.  The authors demonstrate its effectiveness across various scenarios, both in simulations and real-world tasks.  The detailed experiments provide strong evidence of the approach’s robustness and practical utility
- The problem of generating diverse and stable cluttered scenes is well-defined.  The authors articulate the limitations of existing methods clearly, providing a strong motivation for their work.
- The use of reinforcement learning to frame the scene generation task and the introduction of beta distribution for action space seem like significant contributions.  These novel aspects are well-highlighted and effectively demonstrated through the experiments
- The paper shows the practical relevance of ClutterGen through real-world experiments on clutter rearrangement and stable object placement
- The video is fantastic

**Weaknesses**

- One of my primary concerns is that the problem might not be as hard as the authors make it seem.  For instance, I didn’t see any example in the video that considered something like object stacking, and this kind of exploration is not encouraged by the model.  Further, the objects can only be translated and rotated about the z axis, so the envelope of the surface that will touch the table is fixed.  Given this setup, it's unclear why objects would just randomly fall like the pitcher in the video if it's placed on a free part of the table.  Nothing about the pitcher rotating about its z axis on a flat surface should change its dynamics or stability.
- Building on my previous comment, it seems like simple baselines could have been used in the evaluation to further motivate the approach.  For instance, using a potential field approach to avoid collisions and place objects in free spaces is well-established in robotics and seems like a reasonable baseline.  Objects can be guided by repulsive forces from other objects and attractive forces towards the desired position on the table. This method is computationally efficient and intuitive for the problem as described.  Also, treating the problem as a 2D packing problem on the surface in contact with the table could indeed be a simpler and more straightforward approach.  This method is extensively studied and has efficient algorithms that could be adapted for the scenario described.  Now, to be clear, these baselines would not at all be reasonable if full rotation and object stacking would be involved; however, that is not the case in this version of the paper.
- While the authors emphasize the physical simulation to ensure stability, I was surprised to see “positive” examples in the video with large parts of objects hanging off the edge of the surface.  While the simulator may deem this as “physically stable”, allowing for this just seems impractical and unnecessary.  This is, yet again, something that a simpler approach would simply not tolerate and would easily avoid.
- In 3.2: you say “$orientation \in \mathbb{R}^4$”.  However, assuming you are using four values here to represent a unit quaternion, this will not be the space of $\mathbb{R}^4$, this will instead be a Lie group of unit quaternions, commonly referred to as $\mathbb{H}$

**Detailed Assessment**

**Is the problem well explained?**

Partially.  The problem is well-explained and they highlight the significant human engineering effort often required for traditional methods and the need for a more automated and efficient approach.  However, I feel that obvious, simple baselines could easily handle the problem as presented as only translation and z rotation on a flat surface are currently investigated.

**Is the problem  big enough where others in the community will care?**

Partially.  The full version of the problem (with object stacking and full translation and rotation), definitely yes.  The problem is significant and relevant to the robot learning community.  The ability to generate diverse and stable cluttered scenes is crucial for training robotic systems in realistic environments.  However, the version of the problem presented here (only translation and z rotation), I’m not convinced.

**Is the solution well explained?**

The ClutterGen framework is well-explained in detail.  The authors describe their reinforcement learning approach, including the problem formulation, observation and action spaces, policy design, and reward function.  The decision to use beta distribution for policy distribution is particularly well-justified.

**Is the work novel enough?**

The novelty of ClutterGen lies in its approach to framing scene generation as a reinforcement learning problem and using a closed-loop mechanism to ensure scene diversity and stability.  The use of beta distribution for action space is also a notable innovation.  These contributions are well-highlighted and effectively demonstrated through the experiments, establishing the work’s novelty

**Overall assessment**

The paper presents a significant and well-executed contribution to the field of robot learning.  The problem is clearly defined, the solution is well-explained, and the evaluation is robust and comprehensive.  However, the necessity of the complex reinforcement learning approach should be better justified, and comparisons with simpler baseline methods should be included.

**Quality Of The Limitations Section:**

3

**Questions For Rebuttal:**

- The paper should include comparisons with simpler baseline methods like potential fields and 2D packing algorithms.  Demonstrating that ClutterGen significantly outperforms these baselines would strengthen the justification for its complexity.
- Provide a stronger justification for why reinforcement learning is necessary for this problem. This could include a discussion of the limitations of simpler methods in the specific context of ClutterGen’s objectives.
- Address concerns about the perceived complexity of the problem.  Explain why objects might fall in the simulation and how ClutterGen addresses these challenges uniquely compared to simpler methods.
- Longer term, I highly recommend incorporating full translations and rotations.  Many of my previous concerns would go away.

**Robotics Focus:**

4

**Summary Of Paper:**

The paper presents a novel approach to generating physically compliant, cluttered scenes for robot learning using reinforcement learning.  The presented framework framework that casts the scene generation problem as a reinforcement learning task, using a closed-loop mechanism and beta distribution for action space to enhance scene diversity and stability.  The paper provides extensive evaluation through simulations and real-world experiments, demonstrating the effectiveness of ClutterGen in generating diverse and stable object layouts.

**Summary Of Recommendation:**

The paper presents a significant and well-executed contribution to the field of robot learning.  The problem is clearly defined, the solution is well-explained, and the evaluation is robust and comprehensive.  However, the necessity of the complex reinforcement learning approach should be better justified, and comparisons with simpler baseline methods should be included.

---

### Official Review · Reviewer_4LKe · 2024-07-21
**Interesting work and timely given current interests in 3D scene generation**

**Originality:** 4
**Technical Quality:** 5
**Clarity Of Presentation:** 5
**Potential Impact:** 4
**Recommendation:** 4
**Confidence:** 3

**Review:**

**Strengths**
- The paper is well-motivated. With the rapid increase in interest in scalable scene generation, there is a clear need for generating a realistic distribution of ecologically rich scenes. ClutterGen is a great step towards procedural and scalable scene population.
- The analysis experiment is thorough, and effectively highlights the importance of the chosen design decisions in the ClutterGen pipeline.
- The generated diversity is compelling, and showcases ClutterGen's ability to generate diverse arrangements given a fixed set of objects.

**Weaknesses**
- While the efficiency comparison is already compelling, it would be even more convincing to show that an equivalent dataset size _ClutterGen-200_ can either match or even outperform the human-curated equivalent dataset _Human-200_.
- It is suggested that ClutterGen can be applied iteratively to construct a scene graph, e.g.: "fork on top of plate on top of table". It would be powerful to showcase an example of this or something equivalent as a proof-of-concept to show how ClutterGen can fully populate a given scene (e.g.: setting a dinner table).
- I was surprised to see no comparison with a "smarter" rejection sampling technique -- e.g.: in simulation (and presumably in real, given ground-truth object meshes), given a proposed placement pose collision information should be already be known. This coupled with object bounding box information should allow rapid rejection sampling without having to actually step physics many times. Other than speed improvements, what additional key benefits does ClutterGen have over such an approach?

**Quality Of The Limitations Section:**

3

**Questions For Rebuttal:**

Please see Weaknesses section above.
- How were the different dataset groups selected? Were they all randomly selected or was there some semantic / geometric distinction (e.g: "all kitchen items" or "all convex items", etc.)
- One interesting qualitative trait of the learned placement policy is that it appears it is biased towards placing objects towards the edges of the supporting surface (as Fig. 5 also quantitatively shows). Could this possibly bias results and result in object placement distributions that don't necessarily align with "natural" distributions found in the real world? I also wonder if this is in part due to the fact that the policy only observes the current set of already-placed objects and the current queried object instead of the full set of future objects to be placed, leading the policy to avoid placing objects in the center because it limits the possibility for future (unknown) objects to be placed. I am curious to hear the author's thoughts on this or if they have considered this alternative setup!
- Can the method be applied to non-rectangular surfaces, e.g.: a circular or abnormally-shaped table?
- It seemed surprising that the placement policy suffered when the sampling surface was translated (Sec. 4.2). Assuming that the entire surface is still sampleable, I would expect the same placements to be valid. Has any sort of normalization with respect to the sampling surface been considered to prevent overfitting?
- What is the role of the human selecting the preferred setup for the object rearrangement task? It seems that this would mainly allow the user to select setups that are "easier" to achieve for the robot compared to other alternative proposals.
- Minor syntax issues:
	- L147: closet -> closest
	- L148: RSS presumably should be RRS

**Robotics Focus:**

4

**Summary Of Paper:**

The paper presents ClutterGen, an automatic, policy-driven method for procedurally sampling multiple object placements on a given surface. The paper highlights the reliability and scalability of ClutterGen, and showcase applications in downstream robot tasks such as Clutter Rearrangement and Stable Object Placement.

**Summary Of Recommendation:**

The paper presents an interesting method for tackling procedural scene generation, showcasing a learned policy that can automatically place a set of objects stably on a given surface with high success. Given the increasing interest in the broader field of scene generation, this approach may serve as an important step in fully end-to-end methods for ecologically rich scene generation.

---

### Official Review · Reviewer_fQzd · 2024-07-21
**Novel, but not fully proven, approach to important problem of cluttered scene generation**

**Originality:** 3
**Technical Quality:** 3
**Clarity Of Presentation:** 5
**Potential Impact:** 3
**Recommendation:** 3
**Confidence:** 5

**Review:**

Quality: The work is generally high quality. The proposed algorithm is clearly explained and its performance is validated through a variety of interesting experiments. However, important experiments where the work is compared to other recent procedural generation tools from the variety of simulator groups like AI2Thor (ProcThor, Holodeck), BEHAVIOR (iGibson, OmniGibson), etc. are missing - currently it is not obvious whether this method performs better, on par with, or worse than those existing approaches with heuristic rules. Tradeoffs should also be discussed.

Clarity: The work and the presentation are very clear. The problem and the proposed approach are clearly explained. There is not much to comment on.

Originality: The work introduces a fully novel approach to solving a known and important problem (discussed below).

Significance: The problem itself is definitely significant: in simulation environments the lack of realistic clutter is one of the major sources o the so-called sim-to-real gap. The paper tries to solve this by generating scenes with such clutter. But there are a few major issues with the approach:

1. The proposed algorithm provides no way of guiding the generation towards semantically-meaningful clutter rather than just physically stable, and in fact probably captures a very small subset of what's physically stable, too. For example, clutter in a kitchen cabinet or fridge will usually have certain important characteristics about it stemming from the distribution of the objects involved and their placement etc. - simply placing random objects in a rejection sampling fashion during training will NOT be a good approximation of this distribution and thus may not be very meaningful to help robots learn to work in such environments.

2. The proposed algorithm only performs transformations along one rotational axis. This is a major limitation that keeps its generations from being semantically realistic. Whether or not a good policy can be learned in the full three-axis case remains to be seen.

3. The actual merit of the described approach (e.g. RL for clutter generation) can only be understood when the approach is compared to existing heuristic-based generation methods. Such experiments are not included in the manuscript and thus it is hard to gauge the significance of the contribution.

**Quality Of The Limitations Section:**

3

**Questions For Rebuttal:**

1. How do you ensure the diversity / completeness of the physically stable configurations (e.g. doesn't the model just learn to build the _easier_ physically stable configurations rather than a good sample of the full distribution?)

2. How do you ensure semantic realism (e.g. the kind of clutter we see is actually relevant for work a robot might do IRL)?

3. How does the approach compare against clutter generation heuristics from the SOTA simulators rather than just random placement?

**Robotics Focus:**

4

**Summary Of Paper:**

The paper introduces an approach to the problem of cluttered scene generation for robotics simulation: the authors formulate the problem as a MDP and apply RL methods to learn a policy that can place objects in a scene in a physically stable and plausible way.

**Summary Of Recommendation:**

The paper introduces a new approach to an important problem, and is presented well. I believe it should be presented at CoRL despite its shortcomings in terms of the method's limitations and the limited experiments against other solutions.

---

### Author Rebuttal · Authors · 2024-08-05

Please refer to our official comments to each reviewer and the attached Zip file here.

---

### Decision · Program_Chairs · 2024-09-04

**Decision:**

Accept

**Comment:**

Strengths:
1.  Addresses the significant issue of generating ecologically rich scenes for robotic simulation, which is crucial for bridging the sim-to-real gap.

2. Introduces a reinforcement learning (RL) approach to frame the scene generation task, using a closed-loop mechanism and beta distribution for action space.

3. Extensive experiments conducted both in simulations and real-world settings, demonstrating the robustness and practical utility of ClutterGen.

4. The paper is well-written and easy to understand, with detailed explanations of the problem, methodology, and results.

5. Demonstrates the ability to generate diverse and stable object layouts, showcasing practical relevance through real-world experiments.

Weaknesses:
1. The paper does not compare ClutterGen to other recent procedural generation tools (e.g., AI2Thor, BEHAVIOR) or simpler heuristic-based methods, making it difficult to assess its significance and performance relative to existing approaches.

2. The current implementation only performs transformations along one rotational axis, which restricts the semantic realism of the generated scenes.

3. The necessity of using a complex RL approach is not well justified, as simpler methods like potential fields or 2D packing algorithms might achieve similar results for the presented problem.

4. The algorithm focuses on physical stability but lacks mechanisms to ensure semantically meaningful clutter, which is essential for realistic scene generation.

5. The approach does not adequately address more complex spatial constraints or interactions, such as stacking objects or ensuring stability throughout all transformations.